# Polymorphisms in DNA Repair Genes and Association with Rheumatoid Arthritis in a Pilot Study on a Central European Population

**DOI:** 10.3390/ijms24043804

**Published:** 2023-02-14

**Authors:** Grzegorz Galita, Joanna Sarnik, Olga Brzezinska, Tomasz Budlewski, Grzegorz Dragan, Marta Poplawska, Ireneusz Majsterek, Tomasz Poplawski, Joanna S. Makowska

**Affiliations:** 1Department of Clinical Chemistry and Biochemistry, Medical University of Lodz, 92-215 Lodz, Poland; 2Doctoral Study in Molecular Genetics, Cytogenetics and Medical Biophysics, Faculty of Biology and Environmental Protection, University of Lodz, Pomorska 141/143, 90-236 Lodz, Poland; 3Department of Rheumatology, Medical University of Lodz, 92-115 Lodz, Poland; 4Biobank, Department of Immunology and Allergy, Medical University of Lodz, 92-213 Lodz, Poland; 5Department of Pharmaceutical Microbiology and Biochemistry, Medical University of Lodz, 92-215 Lodz, Poland

**Keywords:** rheumatoid arthritis, DNA repair genes, DNA damage response, polymorphisms, SNP

## Abstract

Rheumatoid arthritis (RA) is a chronic, multifactorial autoimmune disease characterized by chronic arthritis, a tendency to develop joint deformities, and involvement of extra-articular tissues. The risk of malignant neoplasms among patients with RA is the subject of ongoing research due to the autoimmune pathogenesis that underlies RA, the common etiology of rheumatic disease and malignancies, and the use of immunomodulatory therapy, which can alter immune system function and thus increase the risk of malignant neoplasms. This risk can also be increased by impaired DNA repair efficiency in individuals with RA, as reported in our recent study. Impaired DNA repair may reflect the variability in the genes that encode DNA repair proteins. The aim of our study was to evaluate the genetic variation in RA within the genes of the DNA damage repair system through base excision repair (BER), nucleotide excision repair (NER), and the double strand break repair system by homologous recombination (HR) and non-homologous end joining (NHEJ). We genotyped a total of 28 polymorphisms in 19 genes encoding DNA repair-related proteins in 100 age- and sex-matched RA patients and healthy subjects from Central Europe (Poland). Polymorphism genotypes were determined using the Taq-man SNP Genotyping Assay. We found an association between the RA occurrence and *rs25487/XRCC1*, *rs7180135/RAD51*, *rs1801321/RAD51*, *rs963917/RAD51B*, *rs963918/RAD51B*, *rs2735383/NBS1*, *rs132774/XRCC6*, *rs207906/XRCC5*, and *rs861539/XRCC3* polymorphisms. Our results suggest that polymorphisms of DNA damage repair genes may play a role in RA pathogenesis and may be considered as potential markers of RA.

## 1. Introduction

Rheumatoid arthritis (RA) is a systemic, chronic inflammatory disease of the joints and surrounding tissues [1]. RA is a serious medical problem, diagnosed annually in 41 people per 100,000 (affecting 1% of the human population) and is the most common autoimmune disorder of the joints [2]. Typically, first symptoms appear at the age of 30–60. RA manifests itself with severe joint pain, stiffness, and swelling, and if not treated properly, leads to disability, chronic pain, and distress. Treatment of RA ameliorates pain and function and improves quality of life; however, a substantial proportion of patients with RA do not achieve remission [3]. The pathogenesis of RA is still unknown, and the main risk factors are genetic, environmental, lifestyle, and hormonal factors [4]. The incidence of RA in identical twins suggests that epigenetic factors may be another risk factor for onset or progression of this disease [5]. RA may have wide heterogeneous manifestations, especially regarding progression rates. The substantial heterogeneity of RA is also present at the molecular level, which is likely to be the cause for the presence of a plethora pathogenic phenotypes. One of the characteristic markers of RA is an inflammation involving initially the synovial membrane of the joints and tendon sheaths, and ultimately the propagation of inflammation [6]. The local inflammation process results in systemic elevations of cytokines and proinflammatory proteins such as C-reactive protein (CRP), atherosclerosis, osteoporosis due to systemic inflammation, or increased risk of developing malignant neoplasms. The synovial membrane is overgrown and infiltrated by cells of the immune system (mostly T and B cells) [7]. RA patients’ lymphocytes are characterized by incomplete activation and lack of response to mitogens. Some of these observations are a consequence of metabolic alternations of lymphocytes. Recent rare evidence (including reports from the authors of this study) suggest that these metabolic alternations are included, but not limited to, the DNA damage response (DDR) [8,9,10,11,12].

DDR is a complex signal transduction pathway and consists of DNA repair pathways and multiple proteins including DNA damage sensors, transducer kinases, and effectors. Repair of DNA damage is carried out through different repair pathways involving numerous protein complexes: direct reversal (DR), three excision pathways—base excision repair (BER), nucleotide excision repair (NER), mismatch repair (MMR)—and two double-stranded break (DSB) repair pathways. The DR pathway is the simplest repair pathway and consists of repair proteins that act directly on a damaged base and reestablish the correct structure without removing the damaged nucleotide. Bulky DNA lesions that induce a structural deformation of the DNA helix are removed by the NER pathway. NER is considered as the primary DNA repair pathway due to removal of a wide variety of other more infrequent DNA lesions. The BER system consists of a wide range of glycosylases that have the ability to recognize and remove various modifications of DNA bases, including methylation, oxidation, and deamination. MMR is involved in the repair of misincorporated bases, small deletions, and insertions. For DSB repair, two main independent pathways are identified: nonhomologous DNA end joining (NHEJ) and homologous recombination (HR). The DDR system is one of the most important cellular mechanisms responsible for the integrity of DNA. Dysregulation of DDR and repair systems is exhibited in various clinical conditions that are associated with cancer susceptibility, accelerated aging, and developmental abnormalities [13].

The hypothesis about the role of DDR in RA onset and/or pathogenesis suggests an increased incidence in patients with RA of some diseases with genetic instability background, such as lymphoma and lung cancer [14,15]. The most common lymphoma in RA patients is diffuse large B-cell lymphoma (DLBCL), whose risk of developing is two to three times higher than in the general population. Recent studies have not confirmed the hypothesis that increased lymphoma risk is associated with increased immunosuppressive treatment in RA patients. As we wrote above, to date, there are only a few reports suggesting the role of DDR in RA onset and pathogenesis. In PBMC (peripheral blood mononuclear cells) isolated from RA patients, there was an increased number of DNA strand breaks and 8-oxoguanine (a general marker for oxidative DNA lesions), which are more sensitive to DNA damaging factors. Moreover, delayed repair of DNA damage induced by ionizing radiation in the PBMC of RA patients has been reported. The reduced expression and activity of one of the key double-strand repair proteins—ATM kinase—has also been demonstrated [8,9,10,11,12]. The disadvantage of these functional studies is the relatively small sample size (up to 10 patients) and superficial analysis, including clinical analysis of RA patients.

Because impaired DNA repair may reflect the variability in the genes that encode DNA repair proteins, it is reasonable to investigate whether variability in the genes coding for DDR proteins may be associated with RA. In the present work, we searched for an association between RA and the variants of single nucleotide polymorphisms (SNPs) of the DDR genes: X-Ray Repair Cross Complementing 1 (*XRCC1*) (*rs25487*), X-Ray Repair Cross Complementing 2 (*XRCC2*) (*rs3218536*), 8-Oxoguanine DNA Glycosylase (*OGG1*) (*rs1052133*), Uracil DNA Glycosylase (UNG) (*rs246079* and *rs151095402*), RAD51 recombinase (*RAD51*) (*rs1801320*, *rs7180135*, *rs1801321*, and *rs2619681*), RAD51 paralog B (*RAD51B*) (*rs963917*, *rs963918*, *rs3784099*, and *rs10483813*), Single-Strand-Selective Monofunctional Uracil-DNA Glycosylase 1 (*SMUG1*) (*rs3087404*), Tumor Protein P53 (*TP53*) (*rs1042522*), RAD52 homolog, DNA repair protein (*RAD52*) (*rs1051669*), Methyl-CpG Binding Domain 4 (*MBD4*) (*rs2307293*), MutY DNA Glycosylase (*MUTYH*) (*rs3219472*, *rs3219489*, and *rs3219493*), MRE11 Homolog, Double Strand Break Repair Nuclease A (*MRE11A*) (*rs2155209*), Nijmegen Breakage Syndrome 1 (*NBS1*) (*rs2735383*), X-Ray Repair Cross Complementing 6 (*XRCC6*) (*rs132774*), X-Ray Repair Cross Complementing 5 (*XRCC5*) (*rs207906*), Protein Kinase, DNA-Activated, Catalytic Subunit (*PRKDC*) (*rs7003908*), Thymine DNA Glycosylase (*TDG*) (*rs4135054*), Exonuclease 1 (*EXO1*) (*rs1776180*), and X-Ray Repair Cross Complementing 3 (*XRCC3*) (*rs861539*). These polymorphisms have been correlated mostly with various tumors, but nothing or little is known about their association with RA.

## 2. Results

### 2.1. Characteristics of the Study Population

The summary of the distributions of selected characteristics of cases and controls is presented in Table 1. There were no significant differences in the distributions of age, sex, and smoking status between cases and controls.

The mean time of disease duration was 19.45 ± 21.36 years. Forty-six patients were currently (for at least one month before blood collection) being treated with methotrexate, six patients with sulfasalazine, and 33 patients had not received disease-modifying anti-rheumatic drugs (DMARDs) within the last month. Glucocorticosteroids (GCS) were used for treatment of 43 patients. Twenty patients did not have rheumatoid factor levels (positive in 80 cases). In addition, the level of inflammation markers (CRP; 14.62 ± 19.57 g/dL) and erythrocyte sedimentation rate (ESR; 26.29 ± 22.26 mm/h) were determined. The disease activity was also assessed based on Disease Activity Score 28-joint count C reactive protein (DAS28)–CRP score (DAS < 2.6, defined as remission and low disease activity: 15 patients; DAS28 above 5.1, as high disease activity: 25 patients). The 60 remaining patients had DAS values between 2.6 and 5.1. All controls had CRP and ESR within normal limits and did not have any chronic disease with inflammatory background.

### 2.2. The Hardy–Weinberg Equilibrium Analysis

Test results for the Hardy–Weinberg (HW) principle are shown in Table 2. Allele distributions of the 15 SNPs were in accordance with the HWE (*p* > 0.05). Five from the remaining 13 SNP (*rs7180135*, *rs2619681*, *rs3219493*, *rs2155209*, and *rs2735383*) were also in accordance with the HW law in the control group. The frequency of the distributions of the remaining 8 analyzed polymorphisms (*rs25487*, *rs3218536*, *rs1801320*, *rs1042522*, *rs861539*, *rs10483813*, *rs1801321*, and *rs1052133*) showed a deviation from the HWE law (*p* < 0.05).

### 2.3. Analysis of a Relationship between the Occurrence of RA and the Studied Polymorphic Variants of DDR Genes

This evaluation was carried out using association studies. These are population-based studies that, in part, allow determination of whether a particular allele of a gene is more common in a group of RA patients than in healthy individuals. For individual polymorphisms, the frequency of each genotype was presented in relation to the presence or absence of RA (Table 3).

Four genetic models were analyzed: codominant, dominant, recessive, and superdominant. The correlation between the RA and codominant, dominant, and recessive models was observed for *rs1801321/RAD51* and *rs963917/RAD51B* polymorphisms. We also found an association between the RA and codominant, dominant, and overdominant models of *rs132774/XRCC6*, *rs207906/XRCC5*, and *rs861539/XRCC3* polymorphisms. *rs7180135/RAD51* and *rs963918/RAD51B* polymorphisms showed correlations with RA in both codominant and recessive models, *rs2735383/NBS1* polymorphism only in the recessive model, and *rs25487/XRCC1* in the codominant model (Table 4).

## 3. Discussion

The heritability of RA varies between populations and is estimated to be about 59% [5]. The genetic background of RA is complex and involves various genetic markers. One main characteristic genetic marker of RA is the human leucocyte antigen (HLA) class II region encoding the HLA-DRB1 protein. It has been well established that the development of RA is associated with the presence of *HLA DRB1 * 03* or *HLA DRB1 * 13*. Other genetic factors include Interferon regulatory factor 5 (*IRF 5*), Signal Transducer and Activator of Transcription 4 (*STAT 4*), Protein Tyrosine Phosphatase Non-Receptor Type 22 (*PTPN22*), Cytotoxic T-Lymphocyte Associated Protein 4 (*CTLA4*), and Tumor Necrosis Factor (*TNF*) gene polymorphisms. The list of genetic factors associated with RA is not complete, and every year novel factors are identified and the genetics of RA continue to evolve. The current study is part of this academic trend.

We observed an association between RA occurrence and polymorphism of the DDR genes. Previous works on identifying polymorphisms of the DDR have mainly focused on SNPs located within BER protein-coding genes. Chen et al. observed that individuals with *rs710079/MPG* and *rs2858056/MPG* SNP may have a higher risk of developing RA [16]. The latter was associated with increased serum level of the MPG protein in RA patients [17]. *MPG* gene encode N-methylpurine-DNA glycosylase of the BER pathway is responsible for removal of alkyl-induced DNA adducts that are associated with genomic instability and carcinogenesis [18]. Likewise, increased serum level of another BER glycosylase, MUTYH, in RA patients was correlated with *rs3219463*. Moreover, the RA patients with this polymorphism were less likely to have arthralgia [19]. Another SNP linked to RA and located within the BER gene (the uracil-DNA glycosylase; *UNG*) is rs246079 [20]. *UNG* excise misincorporated uracil from DNA, preventing the formation of mutagenic U/G mispairs [21]. Polymorphisms of two more key BER genes, *OGG1* and *XRCC1*, have been intensively studied [22,23,24,25]. Both genes are involved in the removal of oxidative DNA lesions. *rs159153/OGG1* and *rs3219008/OGG1* were associated with RA progression among Taiwan’s Han Chinese and Egyptian populations; however, these results were not confirmed in Pakistan and Turkish populations. In contrast, two *XRCC1* (*rs25487* and *rs25489*) gene polymorphisms were associated not only with RA susceptibility but also with the severity of RA in almost all studied, but not Turkish, populations.

The current study slightly differs from previous studies on association of SNP in BER genes with RA. We analyzed ten SNPs in the following BER genes: *XRCC1*, *OGG1*, *UNG*, *SMUG MBD4*, *MUTYH*, and *TDG*. However, no associations with RA were found, except *rs25487/XRCC1*. It seems that this SNP could serve as a genetic marker of RA, independent of population bias. It should be mentioned here that our study is a pilot study—it cannot be excluded that a larger study group would alter the current findings, and the remaining BER SNPs could also be linked to the occurrence of RA. The BER SNPs merit further investigation.

This study appears to be the first to find an association between RA occurrence and NHEJ/HR SNPs. A somewhat remarkable observation was that most of SNPs linked with RA SNPs are located within genes encoding the RecA/Rad51-related protein family, which participates in homologous recombination to maintain chromosome stability and repair DNA damage. An equally striking observation is the association of the occurrence of SNPs within genes encoding two key proteins of the NHEJ system, Ku70 and 80, with RA. Both proteins play a very important role in promoting lymphocyte development. During lymphocyte development, V(D)J recombination occurred. V(D)J is important for the formation of lymphocyte B antigen receptors and heavily depends on somatic hypermutation (SHM). SHM is generated through two distinct phases: DNA cutting with DSB formation by RAG1 and RAG2 proteins and processing of DSB via NHEJ, with Ku70 and Ku80 serving as a biochemical link between the two phases of V(D)J recombination [26]. SNPs located within genes encoding Ku70 and Ku80 could alter/interfere with V(D)J and thus contribute to the pathogenesis of RA as a a loss of tolerance in B cell differentiation, and/or activation in RA is connected with altered V(D)J [27].

## 4. Materials and Methods

### 4.1. Study Groups

The study group included 100 patients with RA (78 women and 22 men; mean age 61.68 ± 13.14 years) selected from patients of the Department of Rheumatology, Medical University of Lodz, and outpatient clinic. A control group of 100 healthy subjects (80 women and 20 men; mean age 48.48 ± 16.04) was recruited from patients without symptoms of chronic inflammatory conditions. This cohort study was approved by the Institutional Bioethics Committee of the Medical University of Lodz (Lodz, Poland) (No. RNN/07/18/KE, approved date: 16 January 2018). All RA patients fulfilled the European League Against Rheumatism/American College of Rheumatology (EULAR/ACR) 2010 diagnostic criteria for RA. We exclude from study persons with past or presence malignancy history in first family degree as a potential reason for DNA instability.

### 4.2. DNA Isolation

Blood samples were collected from both patients and controls into anticoagulant up to 9 mL in tubes. Next, DNA were isolated using GeneMatrix Blood DNA purification Kit (EURx, Gdansk, Poland) according to the manufacturer’s protocol. After isolation, DNA samples were stored at −20 °C in Tris/EDTA (TE) buffer (pH 8.0). DNA concentration and purity were determined spectrophotometrically by measuring absorbance at 260 and 280 nm on Synergy HT spectrophotometr (BioTek, Hong Kong, China).

### 4.3. Determination of Single-Nucleotide Polymorphisms (SNPs)

The frequency of polymorphic variants of genes *XRCC1* (*rs25487*), *XRCC2* (*rs3218536*), *OGG1* (*rs1052133*), *UNG* (*rs246079* and *rs151095402*), *RAD51* (*rs1801320*, *rs7180135*, *rs1801321*, and *rs2619681*), *RAD51B* (*rs963917*, *rs963918*, *rs3784099*, and *rs10483813*), *SMUG1* (*rs3087404*), *TP53* (*rs1042522*), *RAD52* (*rs1051669*), *MBD4* (*rs2307293*), *MUTYH* (*rs3219472*, *rs3219489*, and *rs3219493*), *MRE11A* (*rs2155209*), *NBS1* (*rs2735383*), *XRCC6* (*rs132774*), *XRCC5* (*rs207906*), *PRKDC* (*rs7003908*), *TDG* (*rs4135054*), *EXO1* (*rs1776180)*, and *XRCC3* (*rs861539*) was determined using TaqMan^®^ SNP Genotyping Assays and the TaqMan Universal PCR Master Mix, No UNG (Applied Biosystems, Foster City, CA, USA). The total volume of PCR reaction was 20 μL, including 4 µL 5× HOT FIREPol^®^ Probe qPCR Mix (Solis, Tartu, Estonia), 1 µL DNA (100 ng), 1 ul 20× TaqMan SNP primers, and 14 µL RNA free water. PCR reaction conditions were as follows: polymerase activation (10 min, 95 °C), 30 cycles of denaturation (15 s, 95 °C) annealing/extension (60 s, 60 °C). Genotype determination was made in Bio-Rad CFX96 system (BioRad, CA, USA).

### 4.4. Statistical Analysis

A statistical analysis of allele and genotype frequencies and the Hardy–Weinberg equilibrium was conducted using SNPStats (https://www.snpstats.net/start.htm) [28]. Four genetic models were analyzed: codominant, dominant, recessive, and overdominant. The codominant model is the most common model, which implies that each genotype generates a separate, independent risk. This model combines heterozygous and homozygous genotypes for the variant allele, with homozygous genotypes for the most common allele. The dominant and recessive models assume that the variant allele is sufficiently high to increase the risk of RA. The last model, the super-dominant model, assumes that only the heterozygote contributes to RA risk. The differences in the genotype distributions among RA patients and healthy controls were assessed by the chi-squared test. Results for which *p* < 0.05 were considered statistically significant. The statistical analysis also determined the risk of an event (odds ratio—OR) and the confidence interval (95% CI) with the use of a linear regression model.

## 5. Conclusions and Future Perspectives

In conclusion, our result suggests that the genetic variations within DDR genes may be linked with RA, and these polymorphisms may be a useful additional marker in this disease.

Our study has some limitations. First, our study is a pilot, and further research, performed on a larger group, is needed to establish a correlation between RA and DDR SNPs. Furthermore, no functional work of these nine SNP loci was included. In the future, more efforts are needed to understand the role of these SNPs in the development of RA in the Polish population.

## Figures and Tables

**Table 1 ijms-24-03804-t001:** Distributions of demographic variables between 100 RA cases and 100 controls.

Variable	RA (*n* = 100)	Controls (*n* = 100)	*p* *
Age			
Mean (±SD)	61.68 ± 13.14	48.48 ± 16.04	0.64
Sex Female Male	78 22	80 20	0.82
Smoking			0.83
Never	80	83	
Former	20	17	

RA, Rheumatoid arthritis; * Mann–Whitney rank sum test.

**Table 2 ijms-24-03804-t002:** The Hardy–Weinberg Equilibrium analysis.

Polymorphism/Gene	*p*-Value		
	Totality	Control Group	Study Group
*rs25487/XRCC1*	<0.0001	<0.0001	0.19
*rs3218536/XRCC2*	<0.0001	<0.0001	<0.0001
*rs1052133/OGG1*	0.0023	0.0085	0.089
*rs246079/UNG*	0.4	1	0.23
*rs151095402/UNG*	1	1	1
*rs1801320/RAD51*	<0.0001	<0.0001	<0.0001
*rs7180135/RAD51*	<0.0001	0.05	0.00026
*rs1801321/RAD51*	<0.0001	0.0014	<0.0001
*rs2619681/RAD51*	0.043	0.078	0.3
*rs963917/RAD51B*	0.88	0.84	1
*rs963918/RAD51B*	0.88	0.49	0.43
*rs3784099/RAD51B*	0.38	0.083	0.62
*rs10483813/RAD51B*	0.01	7 × 10^−4^	1
*rs3087404/SMUG1*	0.47	0.23	1
*rs1042522/TP53*	<0.0001	<0.0001	0.083
*rs1051669/RAD52*	0.19	1	0.065
*rs2307293/MBD4*	1	1	1
*rs3219472/MUTYH*	1	0.51	0.53
*rs3219489/MUTYH*	0.11	0.39	0.27
*rs3219493/MUTYH*	0.012	0.11	0.048
*rs2155209/MRE11A*	0.0078	0.31	0.0053
*rs2735383/NBS1*	0.0021	0.37	0.00071
*rs132774/XRCC6*	0.57	0.22	0.044
*rs207906/XRCC5*	0.26	0.002	0.35
*rs7003908/PRKDC*	0.4	0.84	0.16
*rs4135054/TDG*	0.74	1	1
*rs1776180/EXO1*	0.89	0.84	1
*rs861539/XRCC3*	<0.0001	<0.0001	<0.0001

**Table 3 ijms-24-03804-t003:** Basic information and allele frequencies of the 28 selected SNPs.

SNP (Gene Name)	Pathway	Chr *	Positions **	Allele	Minor Allele Frequency	
					Case	Control
*rs25487* (*XRCC1*)	BER	19	43551574	C/T	0.34	0.38
*rs3218536* (*XRCC2*)	DSB	7	152648922	C/T	0.48	0.49
*rs1052133* (*OGG1*)	BER	3	9757089	C/G	0.29	0.26
*rs246079* (*UNG*)	BER	12	109109255	A/G	0.5	0.47
*rs151095402* (*UNG*)	BER	12	109098561	C/T	0.01	0.02
*rs1801320* (*RAD51*)	HR	15	40695330	G/C	0.3	0.24
*rs7180135* (*RAD51*)	HR	15	40731896	A/G	0.48	0.36
*rs1801321* (*RAD51*)	HR	15	40695367	G/T	0.26	0.48
*rs2619681* (*RAD51*)	HR	15	40696823	C/T	0.18	0.22
*rs963917* (*RAD51B*)	HR	14	68595606	A/G	0.28	0.44
*rs963918* (*RAD51B*)	HR	14	68595397	C/T	0.46	0.31
*rs3784099* (*RAD51B*)	HR	14	68283210	A/G	0.27	0.28
*rs10483813* (*RAD51B*)	HR	14	68564567	A/T	0.2	0.22
*rs3087404* (*SMUG1*)	BER	12	54187830	T/C	0.42	0.48
*rs1042522* (*TP53*)	BER/DSB	17	7676154	C/G	0.28	0.33
*rs1051669* (*RAD52*)	DSB	12	913286	C/T	0.2	0.21
*rs2307293* (*MBD4*)	BER	3	129431542	C/G	0.02	0.02
*rs3219472* (*MUTYH*)	BER	1	45338378	C/T	0.19	0.19
*rs3219489* (*MUTYH*)	BER	1	45331833	C/G	0.23	0.22
*rs3219493* (*MUTYH*)	BER	1	45330597	C/G	0.08	0.08
*rs2155209* (*MRE11A*)	DSB	11	94417624	C/T	0.37	0.42
*rs2735383* (*NBS1*)	DSB	8	89935041	C/G	0.3	0.34
*rs132774* (*XRCC6*)	NHEJ	22	41635949	C/G	0.48	0.44
*rs207906* (*XRCC5*)	NHEJ	2	216148178	A/G	0.31	0.2
*rs7003908* (*PRKDC*)	NHEJ	8	47858141	A/C	0.46	0.48
*rs4135054* (*TDG*)	BER	12	103969832	C/T	0.1	0.14
*rs1776180* (*EXO1*)	MMR	1	241848042	C/G	0.44	0.49
*rs861539* (*XRCC3*)	HR	14	103699416	A/G	0.16	0.26

* Chromosome; ** Chromosome position according to the Genome Reference Consortium Human Build 38; BER: Base Excision Repair; MMR: Mismatch Repair; DSB Repair: Double Strand Break Repair; NHEJ: nonhomologous DNA end joining; HR: homologous recombination.

**Table 4 ijms-24-03804-t004:** Correlation of RA with the frequency of genotypes of the DDR genes.

Polymorphism/Gene	Model	Genotype	Control Group	Study Group	OR (95% CI)	*p*-Value
*rs25487/XRCC1*	Codominant	T/T	24 (24%)	15 (15%)	1.00	0.034
		C/T	27 (27%)	39 (39%)	2.31 (1.03–5.20)	
		C/C	49 (49%)	46 (46%)	1.50 (0.70–3.21)	
	Dominant	T/T	24 (24%)	15 (15%)	1.00	0.11
		C/T-C/C	76 (76%)	85 (85%)	1.79 (0.87–3.66)	
	Recessive	T/T-C/T	51 (51%)	54 (54%)	1.00	0.67
		C/C	49 (49%)	46 (46%)	0.89 (0.51–1.54)	
	Overdominant	T/T-C/C	73 (73%)	61 (61%)	1.00	0.071
		C/T	27 (27%)	39 (39%)	1.73 (0.95–3.14)	
*rs3218536/XRCC2*	Codominant	C/C	7 (7%)	4 (4%)	1.00	0.13
		C/T	88 (88%)	95 (95%)	1.89 (0.53–6.68)	
		T/T	5 (5%)	1 (1%)	0.35 (0.03–4.15)	
	Dominant	C/C	7 (7%)	4 (4%)	1.00	0.35
		C/T-T/T	93 (93%)	96 (96%)	1.81 (0.51–6.38)	
	Recessive	C/C-C/T	95 (95%)	99 (99%)	1.00	0.084
		T/T	5 (5%)	1 (1%)	0.19 (0.02–1.67)	
	Overdominant	C/C-T/T	12 (12%)	5 (5%)	1.00	0.072
		C/T	88 (88%)	95 (95%)	2.59 (0.88–7.65)	
*rs1052133/OGG1*	Codominant	C/C	60 (60%)	54 (54%)	1.00	0.64
		C/G	28 (28%)	34 (34%)	1.35 (0.73–2.51)	
		G/G	12 (12%)	12 (12%)	1.11 (0.46–2.68)	
	Dominant	C/C	60 (60%)	54 (54%)	1.00	0.39
		C/G-G/G	40 (40%)	46 (46%)	1.28 (0.73–2.24)	
	Recessive	C/C-C/G	88 (88%)	88 (88%)	1.00	1
		G/G	12 (12%)	12 (12%)	1.00 (0.43–2.35)	
	Overdominant	C/C-G/G	72 (72%)	66 (66%)	1.00	0.36
		C/G	28 (28%)	34 (34%)	1.32 (0.73–2.42)	
*rs246079/UNG*	Codominant	A/A	22 (22%)	21 (21%)	1.00	0.55
		A/G	50 (50%)	57 (57%)	1.19 (0.59–2.43)	
		G/G	28 (28%)	22 (22%)	0.82 (0.36–1.87)	
	Dominant	A/A	22 (22%)	21 (21%)	1.00	0.86
		A/G-G/G	78 (78%)	79 (79%)	1.06 (0.54–2.08)	
	Recessive	A/A-A/G	72 (72%)	78 (78%)	1.00	0.33
		G/G	28 (28%)	22 (22%)	0.73 (0.38–1.38)	
	Overdominant	A/A-G/G	50 (50%)	43 (43%)	1.00	0.32
		A/G	50 (50%)	57 (57%)	1.33 (0.76–2.31)	
*rs151095402/UNG*	---	C/C	97 (97%)	98 (98%)	1.00	0.65
		C/T	3 (3%)	2 (2%)	0.66 (0.11–4.04)	
*rs1801320/RAD51*	Codominant	G/G	70 (70%)	63 (63%)	1.00	0.52
		G/C	13 (13%)	14 (14%)	1.20 (0.52–2.74)	
		C/C	17 (17%)	23 (23%)	1.50 (0.74–3.07)	
	Dominant	G/G	70 (70%)	63 (63%)	1.00	0.29
		G/C-C/C	30 (30%)	37 (37%)	1.37 (0.76–2.47)	
	Recessive	G/G-G/C	83 (83%)	77 (77%)	1.00	0.29
		C/C	17 (17%)	23 (23%)	1.46 (0.72–2.94)	
	Overdominant	G/G-C/C	87 (87%)	86 (86%)	1.00	0.84
		G/C	13 (13%)	14 (14%)	1.09 (0.48–2.45)	
*rs7180135/RAD51*	Codominant	G/G	8 (8%)	13 (13%)	1.00	0.013
		A/G	56 (56%)	69 (69%)	0.76 (0.29–1.96)	
		A/A	36 (36%)	18 (18%)	0.31 (0.11–0.88)	
	Dominant	G/G	8 (8%)	13 (13%)	1.00	0.25
		A/G-A/A	92 (92%)	87 (87%)	0.58 (0.23–1.47)	
	Recessive	G/G-A/G	64 (64%)	82 (82%)	1.00	0.0039
		A/A	36 (36%)	18 (18%)	0.39 (0.20–0.75)	
	Overdominant	G/G-A/A	44 (44%)	31 (31%)	1.00	0.057
		A/G	56 (56%)	69 (69%)	1.75 (0.98–3.12)	
*rs1801321/RAD51*	Codominant	G/G	35 (35%)	15 (15%)	1.00	<0.0001
		G/T	34 (34%)	22 (22%)	1.51 (0.67–3.39)	
		T/T	31 (31%)	63 (63%)	4.74 (2.26–9.96)	
	Dominant	G/G	35 (35%)	15 (15%)	1.00	0.001
		G/T-T/T	65 (65%)	85 (85%)	3.05 (1.54–6.06)	
	Recessive	G/G-G/T	69 (69%)	37 (37%)	1.00	<0.0001
		T/T	31 (31%)	63 (63%)	3.79 (2.11–6.82)	
	Overdominant	G/G-T/T	66 (66%)	78 (78%)	1.00	0.058
		G/T	34 (34%)	22 (22%)	0.55 (0.29–1.03)	
*rs2619681/RAD51*	Codominant	C/C	64 (64%)	69 (69%)	1.00	0.62
		C/T	28 (28%)	26 (26%)	0.86 (0.46–1.62)	
		T/T	8 (8%)	5 (5%)	0.58 (0.18–1.86)	
	Dominant	C/C	64 (64%)	69 (69%)	1.00	0.45
		C/T-T/T	36 (36%)	31 (31%)	0.80 (0.44–1.44)	
	Recessive	C/C-C/T	92 (92%)	95 (95%)	1.00	0.39
		T/T	8 (8%)	5 (5%)	0.61 (0.19–1.92)	
	Overdominant	C/C-T/T	72 (72%)	74 (74%)	1.00	0.75
		C/T	28 (28%)	26 (26%)	0.90 (0.48–1.69)	
*rs963917/RAD51B*	Codominant	G/G	30 (30%)	51 (51%)	1.00	0.0037
		A/G	51 (51%)	41 (41%)	0.47 (0.26–0.87)	
		A/A	19 (19%)	8 (8%)	0.25 (0.10–0.63)	
	Dominant	G/G	30 (30%)	51 (51%)	1.00	0.0024
		A/G-A/A	70 (70%)	49 (49%)	0.41 (0.23–0.74)	
	Recessive	G/G-A/G	81 (81%)	92 (92%)	1.00	0.021
		A/A	19 (19%)	8 (8%)	0.37 (0.15–0.89)	
	Overdominant	G/G-A/A	49 (49%)	59 (59%)	1.00	0.16
		A/G	51 (51%)	41 (41%)	0.67 (0.38–1.17)	
*rs963918/RAD51B*	Codominant	C/C	11 (11%)	19 (19%)	1.00	0.0047
		C/T	40 (40%)	54 (54%)	0.78 (0.33–1.82)	
		T/T	49 (49%)	27 (27%)	0.32 (0.13–0.77)	
	Dominant	C/C	11 (11%)	19 (19%)	1.00	0.11
		C/T-T/T	89 (89%)	81 (81%)	0.53 (0.24–1.17)	
	Recessive	C/C-C/T	51 (51%)	73 (73%)	1.00	0.0013
		T/T	49 (49%)	27 (27%)	0.38 (0.21–0.69)	
	Overdominant	C/C-T/T	60 (60%)	46 (46%)	1.00	0.047
		C/T	40 (40%)	54 (54%)	1.76 (1.00–3.09)	
*rs3784099/RAD51B*	Codominant	G/G	55 (55%)	52 (52%)	1.00	0.2
		A/G	33 (33%)	42 (42%)	1.35 (0.74–2.44)	
		A/A	12 (12%)	6 (6%)	0.53 (0.18–1.51)	
	Dominant	G/G	55 (55%)	52 (52%)	1.00	0.67
		A/G-A/A	45 (45%)	48 (48%)	1.13 (0.65–1.97)	
	Recessive	G/G-A/G	88 (88%)	94 (94%)	1.00	0.13
		A/A	12 (12%)	6 (6%)	0.47 (0.17–1.30)	
	Overdominant	G/G-A/A	67 (67%)	58 (58%)	1.00	0.19
		A/G	33 (33%)	42 (42%)	1.47 (0.83–2.61)	
*rs10483813/RAD51B*	Codominant	T/T	67 (67%)	64 (64%)	1.00	0.07
		A/T	22 (22%)	32 (32%)	1.52 (0.80–2.89)	
		A/A	11 (11%)	4 (4%)	0.38 (0.12–1.26)	
	Dominant	T/T	67 (67%)	64 (64%)	1.00	0.66
		A/T-A/A	33 (33%)	36 (36%)	1.14 (0.64–2.05)	
	Recessive	T/T-A/T	89 (89%)	96 (96%)	1.00	0.056
		A/A	11 (11%)	4 (4%)	0.34 (0.10–1.10)	
	Overdominant	T/T-A/A	78 (78%)	68 (68%)	1.00	0.11
		A/T	22 (22%)	32 (32%)	1.67 (0.89–3.14)	
*rs3087404/SMUG1*	Codominant	T/T	19 (19%)	18 (18%)	1.00	0.28
		T/C	57 (57%)	48 (48%)	0.89 (0.42–1.88)	
		C/C	24 (24%)	34 (34%)	1.50 (0.65–3.43)	
	Dominant	T/T	19 (19%)	18 (18%)	1.00	0.86
		T/C-C/C	81 (81%)	82 (82%)	1.07 (0.52–2.18)	
	Recessive	T/T-T/C	76 (76%)	66 (66%)	1.00	0.12
		C/C	24 (24%)	34 (34%)	1.63 (0.88–3.03)	
	Overdominant	T/T-C/C	43 (43%)	52 (52%)	1.00	0.2
		T/C	57 (57%)	48 (48%)	0.70 (0.40–1.22)	
*rs1042522/TP53*	Codominant	C/C	54 (54%)	55 (55%)	1.00	0.24
		C/G	26 (26%)	33 (33%)	1.25 (0.66–2.36)	
		G/G	20 (20%)	12 (12%)	0.59 (0.26–1.32)	
	Dominant	C/C	54 (54%)	55 (55%)	1.00	0.89
		C/G-G/G	46 (46%)	45 (45%)	0.96 (0.55–1.68)	
	Recessive	C/C-C/G	80 (80%)	88 (88%)	1.00	0.12
		G/G	20 (20%)	12 (12%)	0.55 (0.25–1.19)	
	Overdominant	C/C-G/G	74 (74%)	67 (67%)	1.00	0.28
		C/G	26 (26%)	33 (33%)	1.40 (0.76–2.58)	
*rs1051669/RAD52*	Codominant	C/C	64 (64%)	59 (59%)	1.00	0.22
		C/T	32 (32%)	40 (40%)	1.36 (0.76–2.43)	
		T/T	4 (4%)	1 (1%)	0.27 (0.03–2.49)	
	Dominant	C/C	64 (64%)	59 (59%)	1.00	0.47
		C/T-T/T	36 (36%)	41 (41%)	1.24 (0.70–2.19)	
	Recessive	C/C-C/T	96 (96%)	99 (99%)	1.00	0.16
		T/T	4 (4%)	1 (1%)	0.24 (0.03–2.21)	
	Overdominant	C/C-T/T	68 (68%)	60 (60%)	1.00	0.24
		C/T	32 (32%)	40 (40%)	1.42 (0.79–2.53)	
*rs2307293/MBD4*	---	C/C	97 (97%)	97 (97%)	1.00	1
		C/G	3 (3%)	3 (3%)	1.00 (0.20–5.08)	
*rs3219472/MUTYH*	Codominant	C/C	64 (64%)	65 (65%)	1.00	0.45
		C/T	34 (34%)	30 (30%)	0.87 (0.48–1.58)	
		T/T	2 (2%)	5 (5%)	2.46 (0.46–13.15)	
	Dominant	C/C	64 (64%)	65 (65%)	1.00	0.88
		C/T-T/T	36 (36%)	35 (35%)	0.96 (0.54–1.71)	
	Recessive	C/C-C/T	98 (98%)	95 (95%)	1.00	0.24
		T/T	2 (2%)	5 (5%)	2.58 (0.49–13.62)	
	Overdominant	C/C-T/T	66 (66%)	70 (70%)	1.00	0.54
		C/T	34 (34%)	30 (30%)	0.83 (0.46–1.51)	
*rs3219489/MUTYH*	Codominant	C/C	59 (59%)	57 (57%)	1.00	0.96
		C/G	38 (38%)	40 (40%)	1.09 (0.61–1.93)	
		G/G	3 (3%)	3 (3%)	1.04 (0.20–5.34)	
	Dominant	C/C	59 (59%)	57 (57%)	1.00	0.77
		C/G-G/G	41 (41%)	43 (43%)	1.09 (0.62–1.90)	
	Recessive	C/C-C/G	97 (97%)	97 (97%)	1.00	1
		G/G	3 (3%)	3 (3%)	1.00 (0.20–5.08)	
	Overdominant	C/C-G/G	62 (62%)	60 (60%)	1.00	0.77
		C/G	38 (38%)	40 (40%)	1.09 (0.62–1.92)	
*rs3219493/MUTYH*	Codominant	C/C	86 (86%)	89 (89%)	1.00	0.79
		C/G	12 (12%)	9 (9%)	0.72 (0.29–1.81)	
		G/G	2 (2%)	2 (2%)	0.97 (0.13–7.01)	
	Dominant	C/C	86 (86%)	89 (89%)	1.00	0.52
		C/G-G/G	14 (14%)	11 (11%)	0.76 (0.33–1.76)	
	Recessive	C/C-C/G	98 (98%)	98 (98%)	1.00	1
		G/G	2 (2%)	2 (2%)	1.00 (0.14–7.24)	
	Overdominant	C/C-G/G	88 (88%)	91 (91%)	1.00	0.49
		C/G	12 (12%)	9 (9%)	0.73 (0.29–1.81)	
*rs2155209/MRE11A*	Codominant	T/T	31 (31%)	33 (33%)	1.00	0.19
		C/T	54 (54%)	60 (60%)	1.04 (0.57–1.93)	
		C/C	15 (15%)	7 (7%)	0.44 (0.16–1.22)	
	Dominant	T/T	31 (31%)	33 (33%)	1.00	0.76
		C/T-C/C	69 (69%)	67 (67%)	0.91 (0.50–1.65)	
	Recessive	T/T-C/T	85 (85%)	93 (93%)	1.00	0.068
		C/C	15 (15%)	7 (7%)	0.43 (0.17–1.10)	
	Overdominant	T/T-C/C	46 (46%)	40 (40%)	1.00	0.39
		C/T	54 (54%)	60 (60%)	1.28 (0.73–2.24)	
*rs2735383/NBS1*	Codominant	C/C	41 (41%)	42 (42%)	1.00	0.075
		C/G	50 (50%)	56 (56%)	1.09 (0.62–1.94)	
		G/G	9 (9%)	2 (2%)	0.22 (0.04–1.07)	
	Dominant	C/C	41 (41%)	42 (42%)	1.00	0.89
		C/G-G/G	59 (59%)	58 (58%)	0.96 (0.55–1.68)	
	Recessive	C/C-C/G	91 (91%)	98 (98%)	1.00	0.024
		G/G	9 (9%)	2 (2%)	0.21 (0.04–0.98)	
	Overdominant	C/C-G/G	50 (50%)	44 (44%)	1.00	0.4
		C/G	50 (50%)	56 (56%)	1.27 (0.73–2.22)	
*rs132774/XRCC6*	Codominant	G/G	35 (35%)	21 (21%)	1.00	0.029
		C/G	43 (43%)	61 (61%)	2.36 (1.21–4.61)	
		C/C	22 (22%)	18 (18%)	1.36 (0.60–3.11)	
	Dominant	G/G	35 (35%)	21 (21%)	1.00	0.027
		C/G-C/C	65 (65%)	79 (79%)	2.03 (1.08–3.81)	
	Recessive	G/G-C/G	78 (78%)	82 (82%)	1.00	0.48
		C/C	22 (22%)	18 (18%)	0.78 (0.39–1.56)	
	Overdominant	G/G-C/C	57 (57%)	39 (39%)	1.00	0.011
		C/G	43 (43%)	61 (61%)	2.07 (1.18–3.64)	
*rs207906/XRCC5*	Codominant	G/G	70 (70%)	45 (45%)	1.00	0.0003
		A/G	21 (21%)	48 (48%)	3.56 (1.88–6.71)	
		A/A	9 (9%)	7 (7%)	1.21 (0.42–3.48)	
	Dominant	G/G	70 (70%)	45 (45%)	1.00	0.0003
		A/G-A/A	30 (30%)	55 (55%)	2.85 (1.59–5.10)	
	Recessive	G/G-A/G	91 (91%)	93 (93%)	1.00	0.6
		A/A	9 (9%)	7 (7%)	0.76 (0.27–2.13)	
	Overdominant	G/G-A/A	79 (79%)	52 (52%)	1.00	<0.0001
		A/G	21 (21%)	48 (48%)	3.47 (1.87–6.46)	
*rs7003908/PRKDC*	Codominant	A/A	27 (27%)	26 (26%)	1.00	0.4
		A/C	49 (49%)	57 (57%)	1.21 (0.62–2.34)	
		C/C	24 (24%)	17 (17%)	0.74 (0.32–1.67)	
	Dominant	A/A	27 (27%)	26 (26%)	1.00	0.87
		A/C-C/C	73 (73%)	74 (74%)	1.05 (0.56–1.97)	
	Recessive	A/A-A/C	76 (76%)	83 (83%)	1.00	0.22
		C/C	24 (24%)	17 (17%)	0.65 (0.32–1.30)	
	Overdominant	A/A-C/C	51 (51%)	43 (43%)	1.00	0.26
		A/C	49 (49%)	57 (57%)	1.38 (0.79–2.41)	
*rs4135054/TDG*	Codominant	C/C	74 (74%)	82 (82%)	1.00	0.38
		C/T	24 (24%)	17 (17%)	0.64 (0.32–1.28)	
		T/T	2 (2%)	1 (1%)	0.45 (0.04–5.08)	
	Dominant	C/C	74 (74%)	82 (82%)	1.00	0.17
		C/T-T/T	26 (26%)	18 (18%)	0.62 (0.32–1.23)	
	Recessive	C/C-C/T	98 (98%)	99 (99%)	1.00	0.56
		T/T	2 (2%)	1 (1%)	0.49 (0.04–5.55)	
	Overdominant	C/C-T/T	76 (76%)	83 (83%)	1.00	0.22
		C/T	24 (24%)	17 (17%)	0.65 (0.32–1.30)	
*rs1776180/EXO1*	Codominant	C/C	25 (25%)	31 (31%)	1.00	0.62
		C/G	52 (52%)	49 (49%)	0.76 (0.39–1.46)	
		G/G	23 (23%)	20 (20%)	0.70 (0.32–1.56)	
	Dominant	C/C	25 (25%)	31 (31%)	1.00	0.34
		C/G-G/G	75 (75%)	69 (69%)	0.74 (0.40–1.38)	
	Recessive	C/C-C/G	77 (77%)	80 (80%)	1.00	0.61
		G/G	23 (23%)	20 (20%)	0.84 (0.43–1.65)	
	Overdominant	C/C-G/G	48 (48%)	51 (51%)	1.00	0.67
		C/G	52 (52%)	49 (49%)	0.89 (0.51–1.54)	
*rs861539/XRCC3*	Codominant	G/G	64 (64%)	79 (79%)	1.00	0.048
		A/G	19 (19%)	9 (9%)	0.38 (0.16–0.91)	
		A/A	17 (17%)	12 (12%)	0.57 (0.25–1.28)	
	Dominant	G/G	64 (64%)	79 (79%)	1.00	0.018
		A/G-A/A	36 (36%)	21 (21%)	0.47 (0.25–0.89)	
	Recessive	G/G-A/G	83 (83%)	88 (88%)	1.00	0.31
		A/A	17 (17%)	12 (12%)	0.67 (0.30–1.48)	
	Overdominant	G/G-A/A	81 (81%)	91 (91%)	1.00	0.04
		A/G	19 (19%)	9 (9%)	0.42 (0.18–0.98)	

## Data Availability

The data presented in this study are available on request from the corresponding author. The data are not publicly available due to regulations in the country of the correspondence author.

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
