# Peer review of "Polymorphisms in DNA Repair Genes and Association with Rheumatoid Arthritis in a Pilot Study on a Central European Population"

_ijms, 2023, doi:10.3390/ijms24043804_

Round 1
Reviewer 1 Report
Thank you for your invitation.
I found the article very interesting, and the methodology is correct with some details not included:
Was the only inclusion criterion you used that they had been diagnosed with RA?
How long did it take to recruit the study sample?
Did they take into account the pharmacological treatments followed by the patients in their study sample with RA?
Is the literature up to date?
How did you calculate the study sample?
In your conclusions, you refer [...] The current study slightly differs from previous studies on association SNP in BER genes with RA. [...] (229-230) have they reviewed the current literature? the most recent article in their bibliography is from 2022 (Rheumatoid Arthritis: Pathogenic Roles of Diverse Immune Cells), it is not related to the main topic of the article. do the current genetic lines of research in RA relate to their findings? are there no recently published similar studies?
Author Response
Point: Was the only inclusion criterion you used that they had been diagnosed with RA?
Response: No, we exclude from study persons with past or presence malignancy history in first family degree as a potential reason for DNA instability. This description is in the 2.1 section of our manuscript.
Point: How long did it take to recruit the study sample?
Response: The project started at 2017 so the recruit of the study sample take over 6 years.
Point: Did they take into account the pharmacological treatments followed by the patients in their study sample with RA?
Response: In this manuscript no as the pharmacological treatments has no impact on genetic background. Of course, an inverse relationship is possible albeit the treatment is not yet complete, so we cannot assess it at this stage of the research. This project is being continued, so it is not excluded that in the near future it may be possible to evaluate the influence of genetic background on the effectiveness of the treatment.
Point: Is the literature up to date?
Response : Yes, the literature contains the latest reports on the topics covered in the publication
Point: How did you calculate the study sample?
Response: This is a pilot study, so we did not calculate the study sample. However our study sample did not differ from other, published pilot studies.
Point: In your conclusions, you refer [...] The current study slightly differs from previous studies on association SNP in BER genes with RA. [...] (229-230) have they reviewed the current literature? the most recent article in their bibliography is from 2022 (Rheumatoid Arthritis: Pathogenic Roles of Diverse Immune Cells), it is not related to the main topic of the article. do the current genetic lines of research in RA relate to their findings? are there no recently published similar studies?
Response : Thank you very much for your comments. We have found only a few manuscript regarding only 4 from of the 28 selected for this study SNPs. The remaining 24 have not yet been studied and our research is the first such extensive research in this field. We agreed that this paper did not fit to the main topic of the article, however it contains general information about RA and therefore it deserves to be included in Introduction.
Many thanks for Yours comments
Reviewer 2 Report
In this study, ,,Polymorphisms in DNA repair genes and association with Rheumatoid arthritis in a pilot study on Central European population,, by Grzegorz Galita et al., the authors aimed to evaluate the genetic variation in rheumatoid arthritis within the genes of the DNA damage repair system through base excision repair (BER), nucleotide excision repair (NER), and the double strand break repair system by homologous recombination (HR) and non-homologous end joining (NHEJ).
They genotyped a total of 28 polymorphisms in 19 genes encoding DNA repair-related proteins in 100 age and sex matched RA patients and healthy subjects from Central Europe (Poland).
The results of the study suggest that genetic variations in the DNA damage response (DDR) genes may be related to RA and these polymorphisms may be a useful additional marker in this disease.
The introduction provides sufficient background and includes all relevant references.
The design of the research project is appropriate.
The methods are adequately described.
The results are presented and discussed very well.
The figure and tables are explained in detail.
The final conclusions can be improved.
The manuscript is not overloaded with unnecessary information.
The article has several shortcomings as follows:
- I made some suggested changes in the body of the manuscript;
- double-check abbreviations and make the necessary corrections so that abbreviations are explained when they first appear, both in the abstract and in the text of the manuscript.

Author Response
Point : In this study, ,,Polymorphisms in DNA repair genes and association with Rheumatoid arthritis in a pilot study on Central European population,, by Grzegorz Galita et al., the authors aimed to evaluate the genetic variation in rheumatoid arthritis within the genes of the DNA damage repair system through base excision repair (BER), nucleotide excision repair (NER), and the double strand break repair system by homologous recombination (HR) and non-homologous end joining (NHEJ). They genotyped a total of 28 polymorphisms in 19 genes encoding DNA repair-related proteins in 100 age and sex matched RA patients and healthy subjects from Central Europe (Poland). The results of the study suggest that genetic variations in the DNA damage response (DDR) genes may be related to RA and these polymorphisms may be a useful additional marker in this disease. The introduction provides sufficient background and includes all relevant references. The design of the research project is appropriate. The methods are adequately described. The results are presented and discussed very well. The figure and tables are explained in detail. The final conclusions can be improved. The manuscript is not overloaded with unnecessary information. The article has several shortcomings as follows:
- I made some suggested changes in the body of the manuscript
Response: We are grateful to the Reviewer for the care with which our manuscript was read, for the positive opinion on our article and the appreciation of its scientific value.
Point: - double-check abbreviations and make the necessary corrections so that abbreviations are explained when they first appear, both in the abstract and in the text of the manuscript.
Response : Thank you very much for your comments. We checked and explained the abbreviations when they first appear, both in the abstract and in the text of the manuscript